# OneFit: Unified Neural Garment Simulation using Function-based Representation and Learning

## Abstract

The digital garment modeling using self-supervised learning has significantly evolved in terms of the speed and visual quality of garment deformation simulations. Recent advances have incorporated size-awareness which allows to drape garments realistically, by stretching only to avoid collisions with the human body. It allows their deployment into virtual try-on systems where the goal is to observe garment fitting. However, a major-shortcoming is that they learn mesh-specific models which requires a distinct model to be trained for each mesh representations of a given garment. In this paper, we introduce a novel self-supervised garment simulation approach to learn garment deformations using only functions. First, our PolyFit module converts the garment mesh patches into functions which allows a compact yet detail-preserving representation. Then, OneFit learns the deformations of these patches by restricting the space of the PolyFit function transformations conditioned on different body poses, in a physics-guided and an intrinsic geometry-aware manner. It not only extends to various mesh-representations of a given garment but also to diverse representations of a garment type. Hence, a model trained on single garment can generalise across several garment types. Thanks to its compact representation, it is computationally superior to its counterparts, in terms of both training and inference and scales well to unseen garments. Thus, by training OneFit on a set of garments, a mesh-agnostic, garment-agnostic deformation model can be learnt which can either be finetuned or postprocessed to accommodate unseen garment types. Code will be released upon acceptance.

## 1 Introduction

More than 60% of garments sold online end up in landfills due to their improper fits Duhoux et al. (2024). Designing virtual try-on applications which can allow the users to estimate the best fit can significantly reduce this spill. Traditionally, digital garments are modeled through Physics-based Simulations (PBS) Terzopoulos et al. (1987); Nealen et al. (2006); Narain et al. (2012), which are computationally expensive and therefore, not suitable for real-time applications such as virtual try-on or human animations.

By using garments generated by PBS softwares Nvidia (2018b;a); Software (2018); Designer (2018), supervised learning of garment deformations Gundogdu et al. (2020); Tiwari et al. (2020); Pfaff et al. (2021); Wang et al. (2019a); Zhang et al. (2021); Patel et al. (2020); Corona et al. (2021); Santesteban et al. (2019) was achieved, which allowed a fast inference. However, the tedious process of obtaining large amounts of data through PBS (which still requires manual intervention) coupled with the extended training duration, prohibits the use of these methodologies. Recent advances Bertiche et al. (2021); Santesteban et al. (2022b); Bertiche et al. (2022); Grigorev et al. (2023); Chen et al. (2024) approximate PBS by optimising physical forces contributing to garment motion in an unsupervised fashion. This is a big leap that has reduced the computational workload of garment draping by almost $100\times$. However, much like their supervised counterparts, most of these methods learn a mesh-specific garment model that does not scale to significant changes in mesh topology. Moreover, given that even a simple garment such as tshirt can be diversified with various design changes relating to neck styles, arm and overall length as seen in Fig. 1; training a separate model for each garment is impractical.

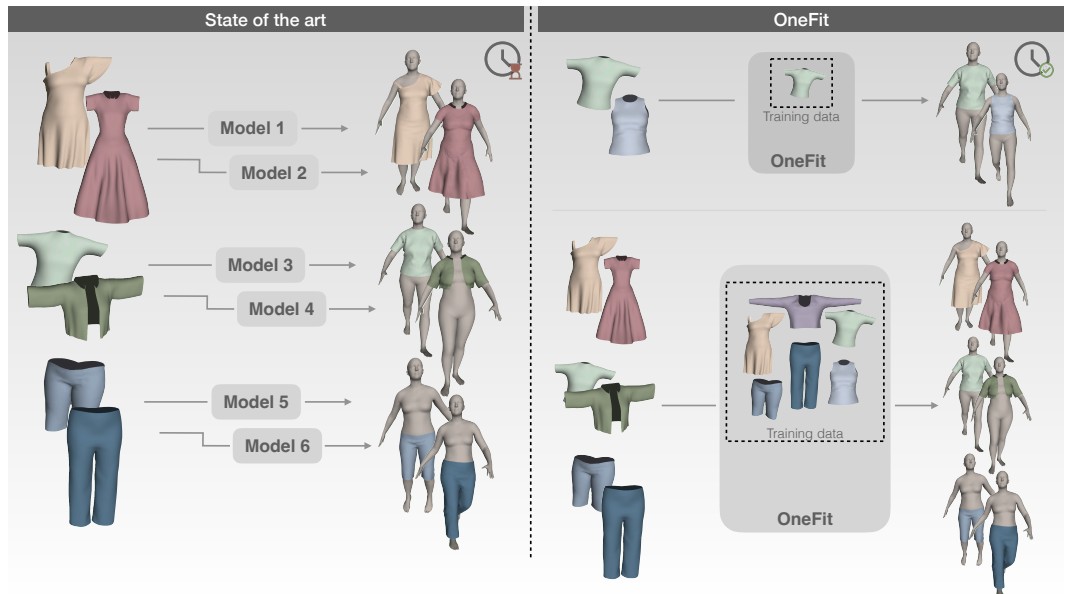

Figure 1: OneFit vs. existing sota methods (supervised or otherwise) . Most methods train a garment-specific (or mesh-specific) model. OneFit, if trained on a single garment (such as half-sleeve tshirt), directly extends to the garments with similar types. It can also learn to drape multiple garments within a single model, resulting in compact models.

In this paper, we present a novel self-supervised draping approach that overcomes the limitation of both mesh-specific and garment-specific learning by adopting a function-based approach for representing garments and learning deformations. We represent the garment as a collection of deformable patches. PolyFit fits a differentiable n-jet function onto each patch in a linear least square sense. It re-orients the patches in order to maximise bijectivity of functions to obtain best possible jet fitting. It is pre-trained on various functions and real garment data. OneFit learns garment deformations as various instances of functions by modifying the PolyFit's jet-coefficients. Instead of using strain and bending forces to model deformations, it conditions surfaces' geometry to deform isometrically (or geodesics-preserving) respecting the tight boundary between patches and garment-body interactions while enforcing physical laws of gravity and body-garment collisions. This local, function-based learning of surfaces allows OneFit to be both mesh-agnostic and garment-agnostic. Consequently, OneFit trained on single garment is able to handle a wide range of inter-class and intra-class garment variations. Moreover, due to its compact function-based representation, it trains quicker than existing methods and provides a much faster inference. Our experiments show that OneFit jointly trained on few garment types ( such as dress, shirt, pants) is able to scale well to a large variety of unseen garment types. Since it does not learn specific body-garment interactions on unseen data, a computationally inexpensive post-processing to remove collision artefacts allows OneFit to drape garments at a minimum of 250 fps, much faster than its counterparts while maintaining a similar drape quality.

## 2 RELATED WORK

**Garment Draping.** Traditional garment simulation methods rely on computationally expensive but accurate differential cloth simulation Narain et al. (2012); Baraff & Witkin (1998); Nealen et al. (2006); Macklin et al. (2016); Liu et al. (2013); Cirio et al. (2014). Advances have been made to reduce the computational complexity of cloth simulation by approximating gradients Li et al. (2022b); Hu et al. (2019) for fast computation or adding 3D priors Guo et al. (2021) such as point clouds of clothed humans. However, these advances compromise reconstruction quality and make the deployment impractical for the virtual try-on systems.

In contrast, learning-based methods yield fast inference. Most methods Bertiche et al. (2020); Patel et al. (2020); Santesteban et al. (2019); Zhang et al. (2021); Wang et al. (2019a); Lähner et al. (2018);

Gundogdu et al. (2020); Guan et al. (2012); Santesteban et al. (2021); Pan et al. (2022); Vidaurre et al. (2020) incorporate a supervised learning approach by using PBS-generated data to learn the relative garment positions with respect to the body. The data generation process is slow and labor intensive which limits the applicability of these methods. Recently, Bertiche et al. (2021); Santesteban et al. (2022b); Bertiche et al. (2022); Chen et al. (2024); Grigorev et al. (2023) proposed unsupervised learning of garment deformations by converting the physical constraints into optimizable losses to estimate garment positions. Most of these methods learn a mesh-specific model which needs to be retrained for slight changes in the garment topology. To our best knowledge, Grigorev et al. (2023) is the only exception that uses graph neural networks to learn drapings of several garment meshes. However, the performance decreases while draping meshes with significantly different resolutions from the training. In contrast, OneFit transforms garment patches into functions to learn drapings of several garments which can handle various mesh resolutions. Moreover, as compared to mesh-based methods, OneFit is less prone to cloth self-intersections.

**Garment representation.** While most learning-based methodologies represent garments as meshes, Zakharkin et al. (2021); Zhang et al. (2023); Bertiche et al. (2020); Ma et al. (2021b) use point based representation to model garment, which allows topologically flexible learning. Ma et al. (2021b) uses dense point cloud to represent garments and obtains a parametric representation using AtlasNet Groueix et al. (2018). Such a global representation is expensive to compute. To ease learning on large point sets with variable sampling resolutions, Ma et al. (2021a) models pose-dependent shape variations of clothing as a collection of rigid patches associated with a set of predefined locations on the body. Ma et al. (2022) builds upon this with a coarse-to-fine prediction of clothing shape to learn highly deformable garments like skirts and dresses. In contrast, OneFit uses deformable patches expressed with simple jet functions using PolyFit which allows an accurate representation of deformable objects.

Some methods Corona et al. (2021); Li et al. (2022a); De Luigi et al. (2023); Santesteban et al. (2022a); Chen et al. (2021); Li et al. (2023) utilize implicit surface functions to handle varying topologies. However, these representations often encounter issues such as self-collisions due to the Signed Distance Function (SDF) inflation and limits to only modeling closed connected surfaces. More advanced representations using Unsigned Distance Function (UDF) De Luigi et al. (2023) can be expensive due to meshing Guillard et al. (2022) and often produce jittery boundaries, which can detract from the realism of the garment simulation. Instead, the localised, explicit, jet fitting of OneFit is computationally inexpensive and accurate.

**Garment Deformation modeling.** While one of the first works on cloth modeling Weil (1986) was purely based on parametric modeling of surfaces using Thin Plate Splines (TPS), physics-based elastic continuum modeling Baraff & Witkin (1998); Liu et al. (2013); Kim (2020); Macklin et al. (2016) combined with collisions, friction, gravity and contact forces is more common in garment simulations. In contrast, Terzopoulos et al. (1987) proposed a purely geometric formulation of modeling deformations by preserving the first and second fundamental forms of the surfaces do Carmo (1976) which allows a direct control on the surfaces' evolution. Most supervised learning-based methods, Bertiche et al. (2020); Ma et al. (2021b); Gundogdu et al. (2020) for example, adopt this scheme and learn deformations by enforcing only geometric constraints (approximated as garment inextensibility and normals similarity) between simulated garment and data used for supervision.

The unsupervised methods Chen et al. (2024); Bertiche et al. (2021) combine physics-based and geometric modeling. While Bertiche et al. (2021) follows a simplistic modeling similar to supervised learning-based methods, Chen et al. (2024) approximates first fundamental form on meshes and enforces an efficient, realistic and tight control which minimizes garment-body collisions. OneFit extends the latter by forcing preservation of the first fundamental form of surfaces.

## 3 ONEFIT

We introduce OneFit, a comprehensive framework that leverages PolyFit ( a patch-wise, function-based representation) to efficiently simulate garment deformations. Figure 2 shows the overview. The template garment $\mathcal{G}_\mathcal{T}$ is upsampled and subdivided using Approximated Centroidal Voronoi Diagrams (ACVD) clustering Valette & Chassery (2004), which efficiently constructs uniform tessellations of a given surface area, into $K$ patches. $K$ varies for each garment. Each patch is passed into

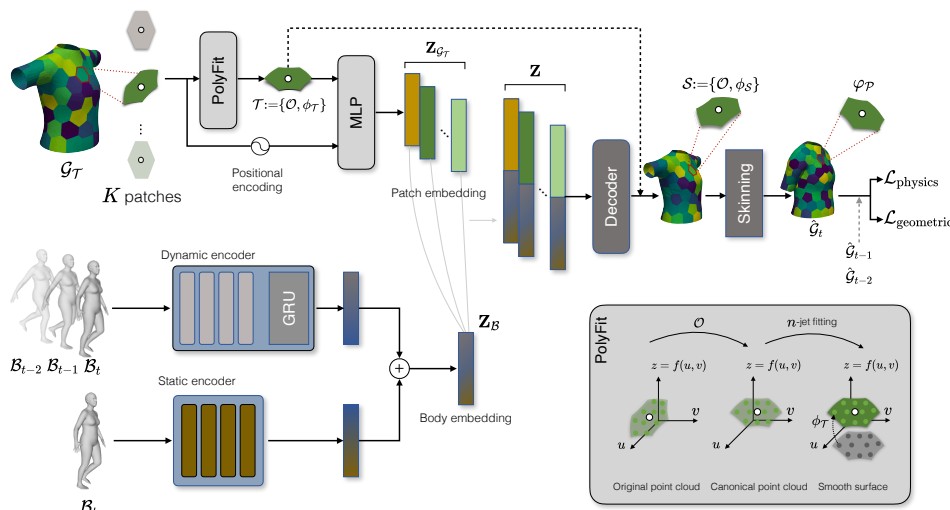

Figure 2: OneFit overview. It divides the garments into small patches. PolyFit fits a parametric function onto each patch which are passed in the downstream to estimate drapings by controlling their geometric and physical behaviour with respect to the body under consideration.

PolyFit which computes orientation $\mathcal{O} = (s, \mathbf{R}, \mathbf{T})$ and a parametric $n$-jet $\phi_{\mathcal{T}}(u, v)$ with respect to a canonical UV space. Thus, we obtain a smooth patch representation, $\mathcal{T} := \{\mathcal{O}, \phi_{\mathcal{T}}(u, v)\}$.

A *garment patch embedding*, $\mathbf{Z}_{\mathcal{G}_{\mathcal{T}}}$, is generated by passing $\mathcal{T}$ for each patch along with its positional encoding into the encoder, an MLP with skip connections. The positional encoding, as described in Mildenhall et al. (2020), is applied to each patch to incorporate its center position and its relative offsets from body joints.

A *body embedding*, $\mathbf{Z}_{\mathcal{B}}$ is obtained as a concatenation of dynamic and static encoding. To describe joint orientation relative to the parent joint, we follow Bertiche et al. (2022) and adopt 6D descriptors Zhou et al. (2019) concatenated with a unit vector with the unposed direction of gravity. This allows to alleviate the discontinuities in the rotation space presented in axis-angle representation. For the structure of the static and dynamic encoder, we adhere to the framework established by Bertiche et al. (2022). The global body pose, $\mathcal{B}(\beta, \theta, \vec{v})$ encapsulates the body shape ($\beta$), the current body pose ($\theta$), and the global velocity of the root joint ($\vec{v}$).

Given $\mathcal{B}(\beta, \theta, \vec{v})$ and $\mathcal{G}_{\mathcal{T}}$, the network first computes the garment patch and body embeddings, $\mathbf{Z}_{\mathcal{G}_{\mathcal{T}}}$ and $\mathbf{Z}_{\mathcal{B}}$ respectively. They are then concatenated and fed into a decoder (details in Appendix A.3) as $\mathbf{Z} = \text{concatenate}(\mathbf{Z}_{\mathcal{G}_{\mathcal{T}}}, \mathbf{Z}_{\mathcal{B}})$ to predict the patch deformations, $\mathcal{S} := \{\mathcal{O}, \phi_{\mathcal{S}}(u, v)\}$. The garment deformations are learnt by enforcing the physical equilibrium of forces and geometric consistency of template and deformed surface patches posed on the desired body after skinning. This enables a self-supervised, mesh-agnostic, garment-agnostic learning of the deformations.

### 3.1 POLYFIT

Following the explicit representation of surfaces in terms of height function, $z = f(u, v)$, from a canonical UV space, an $n^{\text{th}}$ order truncated Taylor expansion of $z$ (also known as $n$-jet), is given by

$$z = f_{\alpha,n}(u, v) = \sum_{i=0}^{n} \sum_{j=0}^{i} \alpha_{i-j,j} u^{i-j} v^j. \tag{1}$$

The combinations of $(\alpha, n)$ allow an analytical representation of various non-trivial geometries, whose $n^{\text{th}}$ order derivatives can be computed precisely. Moreover, given sufficient point samples, $z = f(u, v)$ can be obtained by fitting an $n^{\text{th}}$ order jet in a least square sense Cazals & Pouget (2003). Therefore, canonical representation of surfaces, in which every point is parameterized by a diffeomorphism $\phi_{\mathcal{T}} : (u, v) \mapsto (u, v, f(u, v))^{\top}$, can be oriented (using $\mathcal{O} = \{s, \mathbf{R}, \mathbf{T}\}$) to fit any smooth surface patch embedded in $\mathbb{R}^3$.

Given a set of 3D points $\mathbf{p}$ sampled from a garment patch $\mathcal{T} \in \mathcal{G}_\mathcal{T}$ obtained using ACVD, PolyFit yields a smooth representation $\mathcal{T} := \{\mathcal{O}, \phi_\mathcal{T}(u,v)\}$ such that $\mathbf{p} = s\mathbf{R}\phi_\mathcal{T}(u,v) + \mathbf{T}$. This allows an analytical computation of $n^{\text{th}}$ order (non-trivial) differential quantities on surfaces that will be used to enforce geometric and physics-based constraints on garment deformations.

The inherent arbitrariness of positioning $\mathcal{T} \in \mathcal{G}_\mathcal{T}$ in $\mathbb{R}^3$ can lead to problematic scenarios where $\phi_\mathcal{T}$, exhibits set-valued behavior at some points which violates its bijectivity. To mitigate this issue, we leverage Principal Component Analysis (PCA) to transform each patch into a canonical space of maximally planar patch representations. We hypothesize that it reduces the likelihood of encountering degenerate cases with one-to-many mappings in $\phi_\mathcal{T}$ but it does not ensure its bijectivity. Thus, we incorporate a Spatial Transformer Network (STN) Guerrero et al. (2018) which utilizes quaternion rotations to precisely reorient the patches into a suitable configuration for $n$-jet fitting.

We pre-trained PolyFit on point clouds sampled from regular explicit functions (4-jets, trigonometric, Gaussian and Bessels) and fine-tuned on patches sampled from garment meshes in Cloth3D dataset using ACVD, which enhances its generalizability on various garment topologies. More details related to the training, fitting performance and comparison of PolyFit with other point cloud encoders can be found in Appendix A.1.

### 3.2 GEOMETRIC DEFORMATION MODELING

On a patch $\mathcal{T} \in \mathcal{G}_\mathcal{T}$ seen in Figure 3 with a parametric representation $\mathcal{T} := \{\mathcal{O}, \phi_\mathcal{T}\}$ obtained using PolyFit, any 3D point is given by $\mathbf{x} = s\mathbf{R}\phi_\mathcal{T} + \mathbf{T}$. This patch is deformed to $\mathcal{S} := \{\mathcal{O}, \phi_\mathcal{S}\}$ such that $\mathbf{x} \in \mathcal{S}$ is given by $\mathbf{x} = s\mathbf{R}\phi_\mathcal{S} + \mathbf{T}$. Upon skinning with $\psi_\mathcal{S}$, we obtain $\mathcal{P} \in \hat{\mathcal{G}}_t$ posed on body $\mathcal{B}_t$. We impose patch deformations to be isometric (or geodesics-preserving) and enforce the preservation of their first fundamental form in terms of local metric tensors $\mathbf{g}$ at $\mathcal{T} \in \mathcal{G}_\mathcal{T}$ and $\mathcal{P} \in \hat{\mathcal{G}}_t$. The local metric tensor at $\mathcal{T} \in \mathcal{G}_\mathcal{T}$ is given by $\mathbf{g}_\mathcal{T} = s^2 \mathbf{J}_{\phi_\mathcal{T}}^\top \mathbf{J}_{\phi_\mathcal{T}}$. Upon deformation and skinning, it transforms to $\mathbf{g}_\mathcal{P} = s^2 \mathbf{J}_{\phi_\mathcal{S}}^\top \mathbf{R}^\top \mathbf{J}_{\psi_\mathcal{S}}^\top \mathbf{J}_{\psi_\mathcal{S}} \mathbf{R} \mathbf{J}_{\phi_\mathcal{S}}$. $\mathbf{J}_{\phi_\mathcal{T}}$ and $\mathbf{J}_{\phi_\mathcal{S}}$ can be expressed analytically from the parametric representation obtained in Poly-Fit. $\mathbf{J}_{\psi_\mathcal{S}}$ can be computed analytically from the LBS skinning function Lin et al. (2022).

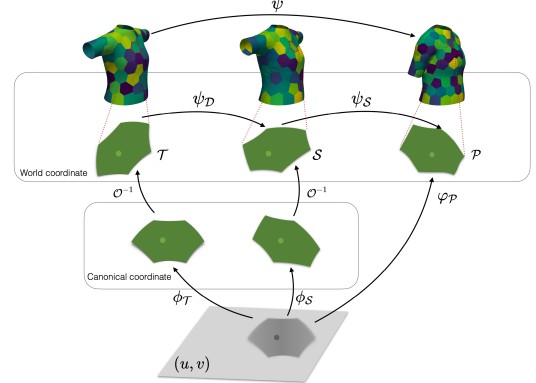

Figure 3: Geometric Deformation Modeling. OneFit deforms $\mathcal{T} \in \mathcal{G}_\mathcal{T}$ isometrically to obtain $\mathcal{P} \in \overline{\mathcal{G}}_t$ posed on body $\mathcal{B}_t$ by forcing patch boundary consistency and avoiding collisions.

Like Chen et al. (2024), we allow local stretchings to avoid collisions. Moreover, we impose geometrical restrictions on patch boundaries to maintain consistency. Thus, we impose the following four geometric losses:

*1) Collision.* It penalizes penetration between the body and the garment. For each points, it is given by

$$\mathcal{L}_{\text{collision}} = k_c \sum_{\text{points}} d_c^2, \tag{2}$$

where $d_c = \max(\epsilon - d(x), 0)$ quantifies the degree of interpenetration. $d(x)$ is the signed distance between garment vertex and body surface, and $\epsilon$ is a small positive constant introduced to enhance stability.

*2) Inextensibility.* In order to preserve geodesic distances between the original and draped garment, it enforces metric tensor similarity. It is computed as

$$\mathcal{L}_{\text{inext}} = k_i \frac{1}{KM} \sum_{\mathcal{T} \in \mathcal{G}_\mathcal{T}} \sum_{\mathbf{x} \in \mathcal{T}} |k_{\text{ext}} \mathbf{g}_\mathcal{T}(\mathbf{x}) - \mathbf{g}_\mathcal{P}(\mathbf{x})| \tag{3}$$

where $\mathbf{g}_{\mathcal{T}}(\cdot)$ and $\mathbf{g}_{\mathcal{P}}(\cdot)$ denote the metric tensor of a point on the template patch and deformed posed patch, respectively. $M$ denotes the number of points in each patch, sampled from the dense mesh vertices and $K$ denotes the number of patches. $k_{\text{ext}} = 1 + \min(d_c, 0.01) \times \min(e, 100)$, where $e$ is the current epoch. We first allow network to stabilise and then enforce inextensibility.

*3) Mesh Inextensibility.* It enforces edge-preserving constraints between the garment and template mesh, $\mathcal{M}_{\mathcal{P}}$ and $\mathcal{M}_{\mathcal{T}}$ respectively .

$$\mathcal{L}_{\text{mesh\_inext}} = k_{\text{mc}} \sum_{i=1}^{n_{\text{edge}}} (e_i(\mathcal{M}_{\mathcal{P}}) - e_i(\mathcal{M}_{\mathcal{T}}))^2 \tag{4}$$

where $e_i(\cdot)$ denotes edge length of the edge $i$-th.

$\mathcal{L}_{\text{mesh\_inext}}$ and $\mathcal{L}_{\text{inext}}$ impose the geodesic preservation constraints at zeroth and first order respectively with points and local jacobians. This allows to restrain the garment deformations to preserve geodesics while taking local body-garment collisions into account.

*4) Boundary.* It enforces the connectivity between adjacent patches and is defined as follows:

$$\mathcal{L}_{\text{boundary}} = \frac{1}{M_b} \sum_{(i,j) \in \mathcal{B}} \sum_{\text{points}} k_{\text{b}} \|\mathbf{x}_i - \mathbf{x}_j\|^2 + k_{\text{bn}} \left(1 - \cos(\theta_n)\right)^2 \tag{5}$$

where $\mathbf{x}_i$ and $\mathbf{x}_j$ denote boundary points on the adjacent patch of index $i$ and $j$, $M_b$ denote the total number of adjacent points between all pairs of patches. $\cos(\theta_n) = \cos\text{\_sim}(\mathbf{N}_0[n], \mathbf{N}_1[n])$ represents the cosine similarity between the normals of the $n$-th pair of adjacent points. This loss effectively penalizes deviations from perfect parallelism between normals, thus promoting smoother transitions at the boundaries.

Overall, the geometric losses are given by

$$\mathcal{L}_{\text{geometric}} = \mathcal{L}_{\text{inext}} + \mathcal{L}_{\text{collision}} + \mathcal{L}_{\text{boundary}} + \mathcal{L}_{\text{mesh\_inext}} \tag{6}$$

## 3.3 PHYSICS-BASED DEFORMATION MODELING

The physics-based losses incorporate effect of interia and gravitational forces. Their implementation is similar to Chen et al. (2024) except they are defined on points instead of mesh vertices.

*1) Gravity.* It incorporates gravity by minimizing the potential energy of the garment, given by

$$\mathcal{L}_{\text{gravity}} = \sum_{\text{vertices}} -mg^\top \mathbf{x}, \tag{7}$$

where $m$ is the particle mass and $g$ is the gravitational acceleration.

*2) Inertia.* It incorporates the inertia loss as proposed in Santesteban et al. (2022b). It is given by

$$\mathcal{L}_{\text{inertia}} = \sum_{\text{vertices}} \frac{1}{2\Delta t^2} m(\mathbf{x}^{[t]} - \mathbf{x}^{[t-1]} - \Delta t v^{[t-1]})^2, \tag{8}$$

where $\Delta t$ is the simulation time step, $\mathbf{x}^{[t]}$ and $\mathbf{x}^{[t-1]}$ specify the particle's position at times $t$ and $t-1$, respectively.

Overall, physics-based losses are

$$\mathcal{L}_{\text{physics}} = \mathcal{L}_{\text{inertia}} + \mathcal{L}_{\text{gravity}} \tag{9}$$

Together, the losses are given by

$$\mathcal{L} = \mathcal{L}_{\text{physics}} + \mathcal{L}_{\text{geometric}} \tag{10}$$

## 4 EXPERIMENTS

### 4.1 IMPLEMENTATION DETAILS

We train **OneFit** on a set of 6 standard garment templates (tshirt, dress, pants, shorts, long-sleeve top and tank) used in garment draping Santesteban et al. (2022a). We utilize the human motion sequences from the AMASS dataset Mahmood et al. (2019) as in Chen et al. (2024); Santesteban et al. (2022a), including 60 sequences with more than 10,000 poses.

We then validate the resulting models on unseen garment meshes from Cloth3D Bertiche et al. (2020), which is a big-scale synthetic dataset containing over 7K sequences of animated 3D humans parameterized using SMPL model and wearing different garments. We note that garments from Cloth3D is first preprocessed to fit on average SMPL body shape in T-pose as described in Appendix A.2.

We set the adaptive batch size according to number of patches of the garment. The learning rate begins at $1e\text{-}3$ for the first 10 epochs and then reduces to $1e\text{-}4$ for the subsequent epochs. Regarding the balancing weights, we set $k_b = 5e3$, $k_{mc} = 2$, $k_g = 1$, $k_c = 1$, and $k_i = 0.5$. These parameters are fixed for all garments across all experiments.

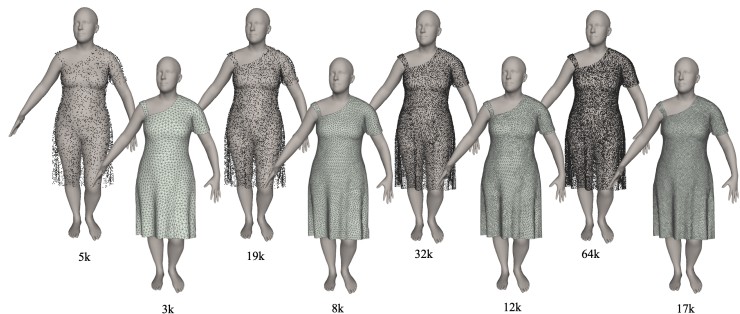

Figure 4: **OneFit** drapings with different mesh resolutions obtained within a similar inference time.

### 4.2 PERFORMANCE EVALUATION

We evaluate the performance of **OneFit** with respect to existing state-of-the-art unsupervised methods: **GAPS** Chen et al. (2024), **SNUG** Santesteban et al. (2022b), **NCS** Bertiche et al. (2022) and **HOOD** Grigorev et al. (2023). Besides **HOOD**, all these methods train mesh-specific, single garment models. **HOOD** trains a mesh-based model but it can train a unified model for multiple garments. **OneFit** trains a mesh-independent model: it can train a single or a multiple garment network. Furthermore, it can finetune an existing model to a specific garment; thus avoiding from-scratch training. Since it learns a mesh-independent model, it can generalise to various mesh resolutions. Figure 4 shows the scalability of **OneFit** towards various mesh resolutions with a similar inference time. **SNUG** and **HOOD** include a post-processing to remove garment-body collision artifacts. **NCS** learns a body-specific model; thus no post-processing is required. **OneFit** does not require post-processing while dealing with garments and bodies in the training dataset or while dealing with garments which cover the garment-body interactions similar to the training data.

**OneFit as a single garment model.** In this experiment, we test the generalization capabilities of our method. Figure 5 shows the results of our method trained on a Tank top. While it drapes well on the trained garment, it generalises well to the garments of similar style without a post-processing. This demonstrates that **OneFit** is highly flexible and generalises well over various garment intra-class variations. As a stress test, we perform another experiment to test the generalisation capabilities of **OneFit** towards garment inter-class variations. Figure 6 (top) shows results of our method trained on a dress and tested on various garments. Since our method learns garment deformations from small patches, it basically learns localised garment-body interactions which are generally extensible to various garments. This is why we see a decent drape on tshirt and tank tops. The only artefacts that appear over these garment are due to collisions. Since the network is learnt on a dress which does not have arms, it has no awareness of the garment-body interactions in this region which makes the collision artefacts inevitable. Given that our method is almost $250\times$ faster than **HOOD** (see timing comparison in Table 5), a simple post-processing can be performed to remove these artefacts with

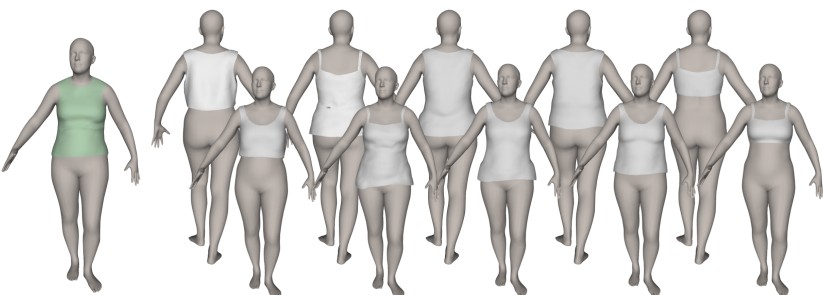

Figure 5: Single garment **OneFit** under garment intra-class variations. Trained on a Tank top (in green), our method is able to drape tank tops of different styles without requiring any post processing.

an inexpensive computation. For more visual details, please refer to the supplementary video. The interesting results in Figure 6 (top) are with pants and shorts which are tightly wrapped to body as compared to dress. Besides the inevitable collision artefacts, some deformation artefacts are also visible within the area between the legs. Since dress is a loose garment, the network does not witness many patches tightly bound between the legs and produces garment artefacts.

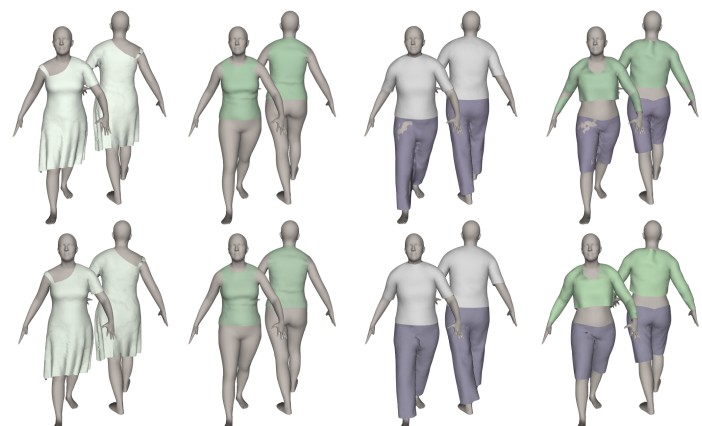

Figure 6: Top: **OneFit** trained using Dress. Bottom: **OneFit** trained using a collection of 6 garments.

Loose garments are known to be challenging for most garment draping methods. Figure 7 shows that our method trained on dress is close with **GAPS**, the best performing method in this case. All other methods yield severe artefacts.

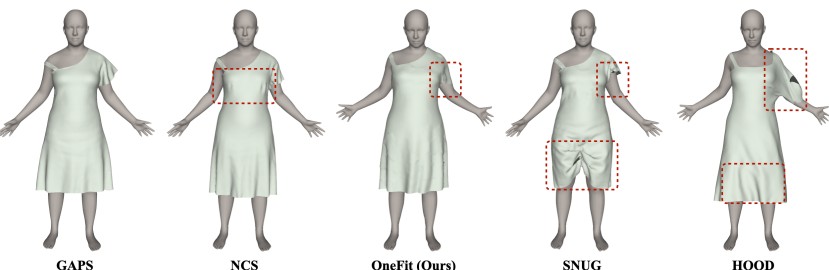

Figure 7: SOTA comparison for **OneFit** trained on dress. The results on **SNUG** and **HOOD** are reported after post-processing to remove collision artefacts.[†]

**OneFit as a multiple garment model.** We train our method jointly on all six garments: tshirt, dress, pants, shorts, long-sleeve top and tank top in order to cover a wide range of body-garment

---

[†]The poses are slightly different due to variations in the SMPL implementation.

| Model | T-shirt | Dress | Tank | Top | Shorts | Pants |
|---|---|---|---|---|---|---|
| **OneFit** (Dress) | 0.330 | 0.840 | 2.834 | 10.033 | 6.271 | 2.389 |
| **OneFit** (6 garments) | 0.422 | 0.756 | 0.481 | 1.592 | 1.749 | 1.194 |

Table 1: $\epsilon_c$ for different configurations. Trained on multiple garment improves the generalizability of the model without requiring any post-processing.

| Model | $\varepsilon_c$ | Training time |
|---|---|---|
| **OneFit** (6 garments) | 2.397 | 8h |
| **OneFit** (6 garments) + finetuning | 1.982 | 1h |
| **OneFit** (jumpsuit) | 1.845 | 3h |

Table 2: Fine-tuning vs training **OneFit** on jumpsuit.

interactions. Table 1 shows that the $\epsilon_c$ has drastically reduced as compared to the inferences made by **OneFit** trained on dress. We have evaluated $\epsilon_c$ on a validation sequence in the AMASS dataset, composing of more than 2,000 frames. Training on multiple garments improves the generalizability of **OneFit**. Figure 6 (bottom) shows that the multiple garment training allows to learn deformations better on pants and shorts; which demonstrated deformation artefacts in Figure 6 (top) under a single garment **OneFit** trained with dress. Figure 8 shows that our method is at par with **GAPS**, the best performing method in this case.

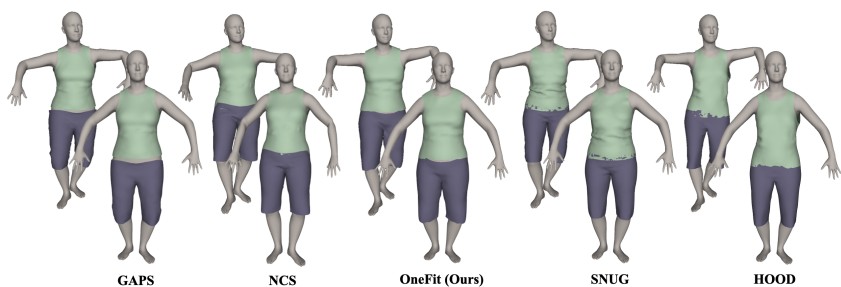

| GAPS | NCS | OneFit (Ours) | SNUG | HOOD |

Figure 8: SOTA comparison on tight garments. The results on **SNUG** and **HOOD** are reported after post-processing to remove collision artefacts.[†]

**Finetuning OneFit.** Once learnt, **OneFit** can be finetuned to a new garment. Table 2 compares the performance of **OneFit** trained on multiple garments to drape a new garment, jumpsuit. Almost 2.5% vertices are observed to be under collision which are brought down to less than 2% by finetuning this model on jumpsuit for an hour. Training **OneFit** from scratch achieves a similar performance with 3× more computation. This allows an fast generalisation of our method to new garments.

**Summary of Experiments.** Since **OneFit** learns garment deformations in terms of local patches, it has high generalizing capabilities. By training **OneFit** on 6 different garments, we have maximised the network's awareness of various localised garment-body interactions. This allows **OneFit** to drape a variety of garments beyond the trained 6. The collision artefacts are common while draping unseen garments with **OneFit**. Given its timing performance, a computationally inexpensive post-processing can be added to remove such collision artefacts. However, in cases where deformation artefacts are observed, the existing **OneFit** can be finetuned to accommodate the new garment.

### 4.3 ABLATION STUDY

**PolyFit.** We conduct a study on the family of parametric functions used for training. Table 3 shows the results. While learning a parametric representation from a single family of functions is still quite accurate, we perform an exhaustive training to minimize the PolyFit errors.

| Function used for training | Height RMSE | Normal Diff (degree) |
|---|---|---|
| Gaussians only | 0.0248 | 5.485 |
| 4 Families | 0.0239 | 5.423 |
| 4 Families + Finetuning with garment patches | 0.0201 | 5.317 |

Table 3: Study on different training data for PolyFit.

**OneFit.** We conduct an ablation study on OneFit's loss components in the table 4, using a tank top as the test garment. Losses $\mathcal{L}_{\text{mesh\_inext}}$ and $\mathcal{L}_{\text{inext}}$ control the stretchability of garment through zeroth-order (point-based) and first-order (normal-based) metrics. $\mathcal{L}_{\text{col}}$ explicitly controls the amount

of body-garment collisions. Omitting $\mathcal{L}_{\text{mesh\_inext}}$ demonstrates that simply incorporating metric tensor inextensibility loss is insufficient. A zeroth-order loss is necessary to control stretching. Without $\mathcal{L}_{\text{inext}}$, forcing inextensibility causes collision artefacts. Excluding $\mathcal{L}_{\text{col}}$ causes more collisions.

| Model | $\varepsilon_e$ | $\varepsilon_a$ | $\varepsilon_c$ |
|---|---|---|---|
| **OneFit** | 7.828 | 13.020 | 0.227 |
| no $\mathcal{L}_{\text{mesh\_inext}}$ | 13.175 | 24.011 | 0.263 |
| no $\mathcal{L}_{\text{inext}}$ | 7.739 | 12.760 | 1.641 |
| no $\mathcal{L}_{\text{col}}$ | 8.004 | 13.373 | 0.387 |

Table 4: Ablation study on variant combinations of loss functions.

| | Train | Runtime |
|---|---|---|
| **SNUG** | 1-8 h | 32.4 ms |
| **HOOD** | 10 h | 125.5 ms |
| **GAPS** | 2-6 h | 5.12 ms |
| **OneFit** | 2-8 h | 0.482 ms |
| + post-processing | - | 4.108 ms |

Table 5: Timing performance.

### 4.4 TIMING COMPARISON

When it comes to training and runtime performance, the mesh specific methods takes less time to train but can not generalize to different topologies. **GAPS** takes 2 hours to converge for tight garments with less than 10k vertices and up to 6 hours for looser garments like dresses. **HOOD** takes 10 hours according to the author. Our method takes 8 hours for training a multiple garment model. The training is carried out on 4 NVIDIA A100 GPUs. As for run-time performance, we measure the processing duration from the stage of raw body pose data to the final garment meshes. For a fair comparison, we use a CMU motion sequence comprising 2,175 frames to evaluate the runtime. The tests are executed on an Quadro RTX 6000. For **SNUG** and **HOOD**, we used the checkpoint and associated script provided by the author. For **GAPS**, we use the author's script for training and prediction. Table 5 shows the comparison. **HOOD** takes greater runtime due to the use of graph neural network and message passing steps. **SNUG** takes less inference time but is slower than **GAPS** because of the additional per-frame collision post-processing. Our approach has the fastest run time performance. For the post-processing, we have reported the maximum time which was observed. In general, it requires 1-2 ms.

**Limitations and future directions.** To our best knowledge, OneFit is the first method to use deformable patches to learn garment simulations. However, several aspects need to be improved. 1) It employs ACVD clustering prior to jet fitting; however, patches with high curvature sometimes do not conform well to jet functions, leading to loss of details. Reducing the patch sizes can fix this issue but it makes the training computationally expensive. An adaptive, curvature-based patch resizing would be optimal to fix this issue. 2) In addition, discontinuity between the patches is noticeable under some extreme poses. A more sophisticated control on boundary is required. We plan to incorporate second-order properties such as curvatures or hierarchical patch representations to fix this issue. 3) We plan to incorporate materials into our formulation in future in order to drape various materials ranging from light to stiff. 4) We plan to devise mechanism to generate more wrinkles to improve realism in the generated dynamics. Overall, the ultra fast inference of OneFit allows one to incorporate simple mechanisms (without retraining) to deal with more complexities: multi-layered garment draping, dealing with non-homogeneous designs including pockets, zippers, buttons etc and dealing with complex garment designs. We plan to address this issue in future.

## 5 CONCLUSIONS

OneFit offers a novel perspective on mesh-agnostic, garment-agnostic, self-supervised learning of garment deformations using functions. By training on patches, the learnt network generalises to various garments. The function-based representation of patches in terms of jet functions obtained using PolyFit, allows an analytical computation of the differential properties of surfaces. This allows a geometrically-consistent, physics-guided learning of deformations that can accommodate a wide range of garments and achieve real-time performance. We contend that OneFit serves as a valuable complement to existing physics-based, self-supervised garment draping techniques. Once trained on a set of garments, it generalised well to wide range of unseen garments. The fast inference allows one to combine OneFit with an inexpensive post-processing to remove the collision artefacts which are observed while draping unseen garments with different body-garment relationships as compared with training data. However unlikely, if deformation artefacts are observed, OneFit can be quickly finetuned to accommodate the unaccounted deformations.

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

# A  APPENDIX

## A.1  POLYFIT: TRAINING PROCESS AND DATASET DETAILS

**Training details.** We use various types of functions to train the rotation correction module in PolyFit. More specifically, we created a dataset consisting of point cloud patches, generated by combining four families of functions, including jet, trigonomtric, Gaussian and Bessel. The dataset comprises 100k patches. The batch size is set to 512, and learning rate is set to 0.001. For every patch, we perform a preprocessing step including normalization, basis extraction and coordinate frame transformation, similar as depicted in Ben-Shabat & Gould (2020). We further fine-tune the model using patches extracted from CLOTH3D training dataset. The intuition behind this rigorous training is to obtain as accurate as possible function-based representation of garment patches.

The four families of functions is defined as follows:

*1) 4-jet:* $f(u, v) = \sum_{i=0}^{4} \sum_{j=0}^{i} \alpha_{i-j,j} u^{i-j} v^j$

*2) Trigonometric:* $T(u, v) = \alpha \cos\left(\theta \sqrt{u^2 + v^2}\right)$

*3) Gaussian:* $G(u, v) = \alpha \exp\left(-\frac{(u-u_0)^2 + (v-v_0)^2}{2\sigma^2}\right)$

*4) Bessel:* $B(u, v) = \alpha J_0\left(k\sqrt{(u - u_0)^2 + (v - v_0)^2}\right)$

where $\alpha \in [-0.5, 0.5]$, $\theta \in [\pi, 2\pi]$, $\sigma \in [0.5, 1]$ and $k = 5$. Here, $J_0$ denotes the Bessel function of the first kind of order 0. Using $(u, v) \in [-1, 1]$, we sum the outputs from the four functions and train the PolyFit model in an unsupervised way, by minimizing the height discrepancies between the original and the fitted surface points.

**Fitting performance.** To evaluate the fitting performance of PolyFit, we use garment models from the Cloth3D validation dataset Bertiche et al. (2020). Specifically, we extracted 100k patches from up-sampled mesh and compute ground truth normals from their corresponding meshes. The trained PolyFit model is then fine-tuned with these patches to improve its efficacy. We compute its performance from metrics including height RMSE and normal loss, measured in degrees. Figure 9 shows the performance of n-jet fitting on the Cloth3D dataset. This shows that the 4-jet function is capable of fitting point clouds from garment patches effectively. Therefore, we fix $n = 4$, as this setting has been shown to achieve accuracy on garments with reasonable computational complexity.

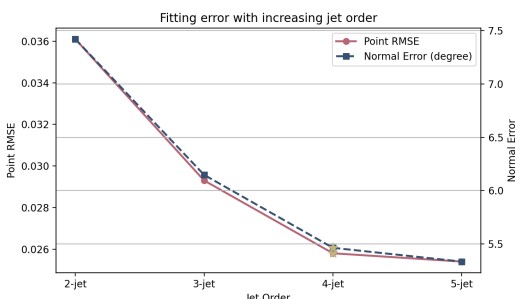

Figure 9: Fitting error for patches from the Cloth3D dataset

Table 6 indicates that the QSTN network noticeably enhances the model's fitting accuracy as it re-orients patches to improve their bijectivity, which leads to better jet-fitting. Additionally, we show PolyFit's performance on six garments by calculating both the point RMSE (Root Mean Square Error) and the normal errors in degrees. These metrics measure the discrepancies between the original upsampled mesh vertices and the corresponding points fitted using PolyFit. The results are provided in Table 7.

|  | height RMSE | normal diff (degree) |
|---|---|---|
| with | 0.0201 | 5.3170 |
| w/o | 0.0259 | 5.4651 |

Table 6: PolyFit fitting metric, with and w/o QSTN.

**PolyFit vs Point cloud encoders.** We evaluated the fitting performance using three methods on patches from Cloth3d validation dataset Bertiche et al. (2020): PolyFit, a PointNet encoder, and a

|                           | Tshirt      | Dress       | Tank        | Top         | Shorts      | Pants       |
|---------------------------|-------------|-------------|-------------|-------------|-------------|-------------|
| Point RMSE                | $8.645e$-05 | $5.303e$-04 | $1.565e$-04 | $2.291e$-04 | $8.005e$-04 | $7.957e$-04 |
| Avg. normal error (degree) | 1.834       | 4.273       | 2.666       | 2.951       | 6.309       | 3.193       |

Table 7: PolyFit performance on garments.

DGCNN encoder, all trained in unsupervised fashion. The metrics used for comparison are height RMSE and normal difference (in degrees).

| Model                     | Height RMSE | Normal Diff (degree) | Avg Inference Time per Patch (ms) |
|---------------------------|-------------|----------------------|-----------------------------------|
| PolyFit                   | 0.0201      | 5.274                | 0.0481                            |
| PointNet Qi et al. (2017) | 0.0309      | 6.936                | 0.0754                            |
| DGCNN Wang et al. (2019b) | 0.0290      | 6.406                | 0.0625                            |

Table 8: Study on different training data for PolyFit.

The results presented in table 8 demonstrate that PolyFit provides slightly superior fitting performance in terms of both accuracy and efficiency. Beyond its marginally superior accuracy, the main reason to use PolyFit is to leverage its bijective function representation to avoid self-intersections and its compact representation. Using PolyFit, a given patch can be represented using $\frac{n(n+1)}{2}$ jet-coefficients (n is the jet order) which is significantly lower than other representations. The fast training and inference time of OneFit are attributed to this compact representation.

## A.2 GARMENT PREPROCESSING

**Patch division.** The garment mesh is first subdivided four times to achieve a dense mesh. Subsequently, ACVD is applied to the refined mesh, clustering the vertices into $n$ patches according to the superficial area. Specifically, the number of patches is given by $\max\left(100, \min\left(400, \lfloor\frac{A}{0.008}\rfloor\right)\right)$, where $A$ denotes the area of the mesh.

**T-pose average shape conversion.** The garments in Cloth3D dataset are with legs slightly separated, which varies from standard T-pose on which skinning weight is computed. Furthermore, the dataset are fit on different body shapes. To test the garment from Cloth3D with OneFit, the garment is first preprocessed to fit average body shape under standard T-pose. We query the closest body vertex for each garment vertex, and then move it according to the displacement of the body vertex between the original and the standard body. Laplacian surface smoothing of single iteration is applied subsequently to smooth the surface. For loose garment including dress and skirt, since they do not adhere to the legs, we only correct the position in terms of shape difference.

## A.3 ONEFIT: NETWORK AND TRAINING DETAILS

In the Dynamic encoder, different from Bertiche et al. (2022), the Gated Recurrent Unit (GRU) layers are initialized with random hidden states. The body feature extractor are implemented using a five-layer multilayer perceptron (MLP) with LeakyReLU activation between the layers. Each layer contains 256 nodes, with the exception of the final layer.

The decoder consists of four fully connected layers, each with dimensions of 512, 512, 512, and 256, respectively. This is followed by three prediction heads for jet coefficient, translation and scale, each implemented as a three fully connected layers with dimensions 128 and 64, ending with a final output layer.

Finally, to maximize parallel computation on GPUs, the batch size for each garment is dynamically determined based on the number of patches using the following equation: $bs = \frac{20,000}{\text{number of patches}}$.

