# OpenReview forum: "OneFit: Unified Neural Garment Simulation using Function-based Representation and Learning"
_ICLR.cc/2025/Conference — ICLR 2025 Conference Withdrawn Submission_

### Official Review · Reviewer_z7Xo · 2024-11-01

**Soundness:** 3
**Presentation:** 3
**Contribution:** 3
**Rating:** 8
**Confidence:** 3

**Summary:**

The paper propose a framework called OneFit for learning garment deformation. Unlike traditional methods that require training a separate model for each garment type, OneFit is trained on a few types of garments and can generalize to previously unseen garment meshes within a single model.

The framework operates by dividing the garment mesh into patches and approximating each patch using a Taylor series expansion. These patches are then processed by a multi-layer perceptron (MLP) to obtain garment embeddings. Simultaneously, body movement and shape are represented through dynamic and static descriptors to form a body embedding. The garment and body embeddings are concatenated to predict garment deformation in response to body movements. To ensure realistic and physically plausible deformations, the training process incorporates both geometric and physics-based loss functions.

**Strengths:**

-  A key innovation in this work is the use of patches to represent garment meshes. By segmenting the mesh into patches using Voronoi diagrams, the authors introduce a structured approach that allows for more flexible and detailed modeling of garment surfaces.

- Each patch is approximated by polynomial function using a fourth-order Taylor expansion, which enables the model to capture intricate geometric information, including curvature and fine wrinkle details. This level of approximation is a noteworthy contribution as it allows for more realistic garment deformations and adds fidelity to the representation of garment.

**Weaknesses:**

- Given that the primary novelty of the paper lies in representing each patch with a polynomial function using Taylor expansion, it would be valuable to see a comparison of different polynomial orders. Such an analysis could provide insights into how varying the order affects the level of detail captured, especially in complex areas with fine geometric details like wrinkles and curvatures.

- The paper does not thoroughly explore the impact of patch resolution on model performance. Since smaller patches may simplify the polynomial approximation, an analysis of the number and size of patches would be beneficial. This would clarify how patch resolution influences model accuracy and computational efficiency.

- The approach to positional encoding remains unclear, especially given the use of multiple garment types for training. Details on how positional encodings are assigned for each garment would be helpful to assess if they are consistent across garments or if garment-specific encodings are used.

**Questions:**

- How does the choice of polynomial order impact the quality of garment deformation details? It would be helpful to understand whether different polynomial orders were tested and how each affects the model's ability to capture detailed garment features.

- How was the patch resolution determined, and were different resolutions tested? Since the size and number of patches may influence both model complexity and accuracy, insight into this decision process would be valuable. Did the authors conduct any experiments to determine an optimal balance between patch resolution and model performance?

- Could you clarify the positional encoding strategy? Specifically, is positional encoding applied globally across all garment types, or is it localized per garment type with unique indices for each? Understanding this would help assess the model's ability to generalize across diverse garment types.

---

### Official Review · Reviewer_C5Ld · 2024-11-04

**Soundness:** 1
**Presentation:** 1
**Contribution:** 2
**Rating:** 3
**Confidence:** 4

**Summary:**

This paper proposes to use deformable patches to simulate various garments and adopt MLP and Transformers as the model architecture. Experiments show lower errors on different garments.

**Strengths:**

* The design is agnostic to the topology of garments.
* The training and inference speed is faster.

**Weaknesses:**

1. Several variables in the equations and the metric in experiments are unclear. What does the $k$ mean in Equation 2,3,4,5? What does the $\epsilon$ mean in Table 1? The explanations are hard to find. Moreover, the meaning is physics loss in Equation 9 is invalid. In SNUG and HOOD, the physics loss is equivalent to solving an optimisation problem defined by the potential energy and external forces, where the potential energy indicates the internal forces in physics. However, Equation 9 itself is incomplete and violates the optimisation problem, indicating invalid use of loss term. Finally, the proposed method seems unable to deal with different materials of garments.
2. The comparisons with baselines are insufficient and less convincing.
    1. Only limited qualitative results are displayed in Figure 7, which only includes one pose of the human. Since this is about garment animation, at least few images indicating the dynamics should be provided. Though there are several animations in the supplementary video, the baselines are not compared simultaneously with the proposed method, i.e. each time only comparing one garment sequence with one baseline instead of comparing one garment with all baselines.
    2. The results related to HOOD are questionable. Firstly, the training time for HOOD mentioned around L502 is 10 hours, while in the original paper the time is 26 hours on single GPU. On the other hand, the 8 hours achieved by proposed method is tested on 4 GPUs, which is unfair to compare with HOOD. How long will it take to train HOOD 4 GPUs? Secondly, around L363, the author claims that HOOD need post-processing to remove the collisions. However, the collision artifacts are well solved through the collision loss in the original paper. Even for results on unseen cases, HOOD shows less collisions without post-processing, but the proposed method need extra post-processing on unseen garments to remove the artifacts, suggesting limited generalisation abilities and robustness.
    3. No quantitative results comparing with baselines are provided. The only quantitative results in Table 4 do not include baselines mentioned in this paper. Based on the weaknesses mentioned above, the results are less convincing and insufficient to verify the effectiveness of the proposed method.
3. As mentioned at L513, the author claims that this is the first method to use patches for simulations. However, LayersNet [1] also adopts patches to learn garments with Transformer-based model, which is also agnostic to garment topologies and very close to the settings in this paper. The major difference is that LayersNet did not use physics loss during training. The author should discuss or even compare with LayersNet using similar training loss to verify the effectiveness of the proposed method.
4. The deformations in the supplementary materials tend to be rigid, especially the end of the dress with fixed wrinkles.

In summary, this paper could be better polished and not ready yet.

[1]. Yidi Shao, et al. Towards Multi-Layered 3D Garments Animation, ICCV2023.

**Questions:**

Please refer to the weaknesses.

---

### Official Review · Reviewer_ScBe · 2024-11-05

**Soundness:** 1
**Presentation:** 2
**Contribution:** 1
**Rating:** 3
**Confidence:** 4

**Summary:**

This paper presents a self-supervised garment simulation approach that bypasses mesh-specific limitations by using a function-based representation. The PolyFit module converts garment mesh patches into a compact form that preserves geometric detail, allowing OneFit to generalize across different garment meshes and styles. By conditioning localized patches on body poses, OneFit achieves mesh-agnostic and garment-adaptive deformations.

**Strengths:**

The approach of using patchwise function for representation in this paper is interesting as it enables the model to handle garment deformations in a flexible, mesh-agnostic manner, and there are some experiments showing its benefit of being faster and generalizing better.  The presentation of this concept is clear, with the authors providing structured explanations and visual diagrams that effectively convey how the PolyFit module supports this localized, function-based learning approach.  Experiments are generally well presented with details elaborated.

**Weaknesses:**

A primary concern with the paper is that the physics-based dynamics appear inaccurate, performing worse than the state of the art. Specifically, the model produces garment sequences that look "tight fitting" and too closely follow the motion of the character, which is a common issue in methods relying on linear blend skinning. This limitation is evident in the supplementary video, where the tops and dress exhibit movement that seems rigidly attached to the character’s motion, lacking the expected flow and lag typical of realistic fabric behavior. Additionally, the paper lacks quantitative comparisons to ground truth sequences, which could illustrate these issues more clearly, particularly for looser garments like dresses. Furthermore, the absence of experimental results comparing the geometric and physics losses with those of other methods and ground truth data limits the assessment of the method's physical accuracy and overall performance.

**Questions:**

1. Handling of Dynamic Behaviors: Could the authors clarify how the method is designed to handle the dynamic behaviors of garments, especially for loose-fitting items like dresses? The results in the supplementary video suggest that the garments move too closely with the character’s motion, giving a "tight-fitting" appearance. Could the authors provide more details on how they might improve or adjust the model to better capture the expected fabric lag and flow?

2. Comparisons: First, the paper lacks quantitative and qualitative comparisons to ground truth sequences, particularly for dynamic accuracy. Second, could the authors include or discuss quantitative evaluations against ground truth or other state-of-the-art models, especially on metrics like geometric invariances, physical accuracy, and external and self collisions?

---

### Official Review · Reviewer_t1Ug · 2024-11-06

**Soundness:** 2
**Presentation:** 2
**Contribution:** 1
**Rating:** 1
**Confidence:** 5

**Summary:**

This paper claims to present a novel self-supervised draping approach that overcomes the limitation of both mesh-specific and garment-specific learning. It further claims to be one trained on single garment and is able to handle a wide range of inter-class and
intra-class garment variations.

However, this paper misses some critical works that already does what this paper claims and missed comparison with those methods.

**Strengths:**

Please see the weakness section.

**Weaknesses:**

### The two major notable claims of this paper are :
- Claim 1. Train on one garment and it will generalize to garment of other types
- Claim 2. Handle various mesh resolutions

**Both the above claims are incorrect, and the solutions for these were also proposed by below papers**
- _Claim 1 & 2_:  GarSim WACV 2023, proposed a unified garment simulator that can *simultaneously learns* deformation of *multiple type of garments (e.g., tops, skirts etc.) of varying topologies and fabrics* conditioned on the underlying body shapes, pose and motion . It is also shown that *GarSim can be trained and tested on the arbitrary resolution garment meshes*. The GarSim is based on graph neural network, hence the statements in the lines 115-120 are also incorrect.
- _Claim 1 & 2_: A similar follow-up work called GenSim CVPR 2023, also proposed a generic method for unsupervised 3D garment simulator that can be *trained simultaneously for multiple types of garments of varying sizes, and topology, and bodies of different shapes, poses and resolutions*. Generalizing on unseen garments of different types, and sizes, along with different body shapes and poses.

- Additionally, both GarSim and GenSim predicts the fabric aware garment deformation. Which is one of the limitations of the OneFit as mentioned in the limitation section.  I encourage the authors to look at the Table 1 of both GenSim and GarSim papers and pitch your claims accordingly.

### The video results shown in the supplementary youtube video are not very encouraging and has flaws.
- In general, one all garments are body hugging and short, not sufficient to validate the claims of the paper. The one large skirt animation is also not realistic, garment appears stiff, as body moves, as we can see while jumping and twisting. Better results for the similar garments have been shown by the HOOD, SNUG, GarSim in their supplementary.
- Timestamp 0.36, the without post-process result has heavy collision with the body. This indicates the collision loss is not working properly. It may be the case it is working fine with the body-hugging garments to some extent. But from results, it appears drastically failing.
- Timestamp 1.06 onwards, the movement of the garments are very siff and unrealistic, a loose garment like skirt should not strictly follow the underlying body movement, it should free flow under the movement of body and gravity.

The authors have missed two important papers to discuss and consider in this research. Which have proposed solutions to the problem setting posed by the onefit paper, the OneFit paper therefore is incomplete and have very limited novelty. Also, the video results are not very encouraging as it has very unrealistic animations and working ok only for body hugging garments only.

Because of the above reasons, I am giving my rating. And encourage the authors to re-write the OneFit paper considering the solutions proposed by GarSim and GenSim. Perhaps mentioning their limitations and how onefit overcomes them. A detailed comparison with them is must if the claims are similar to what they have made in their papers.

**Questions:**

Please see the weakness section.

---

### Note · Authors · 2024-11-26

I have read and agree with the venue's withdrawal policy on behalf of myself and my co-authors.